# FINDING STRUCTURE IN CONTINUAL LEARNING

## ABSTRACT

Learning from a stream of tasks usually pits plasticity against stability: acquiring new knowledge often causes catastrophic forgetting of past information. Most methods address this by summing competing loss terms, creating gradient conflicts that are managed with complex and often inefficient strategies such as external memory replay or parameter regularization. We propose a reformulation of the continual learning objective using Douglas-Rachford Splitting (DRS). This reframes the learning process not as a direct trade-off, but as a negotiation between two decoupled objectives: one promoting plasticity for new tasks and the other enforcing stability of old knowledge. By iteratively finding a consensus through their proximal operators, DRS provides a more principled and stable learning dynamic. Our approach achieves an efficient balance between stability and plasticity without the need for auxiliary modules or complex add-ons, providing a simpler yet more powerful paradigm for continual learning systems.

## 1 INTRODUCTION

Continual learning (CL) aims to train models on a sequence of tasks, emulating human-like learning, but is fundamentally constrained by the stability-plasticity dilemma (French, 1999; Knoblauch et al., 2020). Models must be plastic enough to acquire new knowledge yet stable enough to retain prior knowledge, avoiding catastrophic forgetting of past tasks (Thapa & Li, 2024; Bonnet et al., 2025; Shen et al., 2024). Standard CL methods address this by adding a regularization term to the task loss, $\mathcal{L}_{\text{CL}} = L_{\text{new-task}} + R_{\text{regularization}}$. This coupling forces stability and plasticity into direct competition: stronger regularization slows adaptation, while weaker regularization accelerates forgetting (Elsayed & Mahmood, 2024; Yoo et al., 2024). The most successful approaches are often complex workarounds. Replay methods mitigate forgetting by storing past data but at the cost of significant memory growth (Wu et al., 2024; Yoo et al., 2024; Elsayed & Mahmood, 2024; Thapa & Li, 2024; Eskandar et al., 2025). Architecture-based methods (Rusu et al., 2016; Konishi et al., 2023; Lyle et al., 2024) isolate knowledge by adding new components for each task, leading to unsustainable model growth and restricting knowledge transfer. It's like buying a new bookshelf for every new book rather than learning how to organize them on one. These approaches focus on preventing damage to prior knowledge rather than leveraging it to accelerate new learning. We argue that the core issue lies not in the objectives themselves, but in the optimization strategy that forces them into a direct tug-of-war (Polson et al., 2015; Feng et al., 2022; Bian et al., 2024). The solution, therefore, is not to simply balance this conflict, but to change the nature of the interaction. Instead of modifying the model architecture or adding complex components like memory buffers, we offer a new insight into stability and plasticity objectives through the lens of operator splitting techniques. We employ Douglas-Rachford Splitting (DRS) (Gabay & Mercier, 1976), a powerful algorithm that reformulates the optimization of the task-fitting term ($f$) and the stability term ($g$) into a structured negotiation. Under this formulation, a CL update would no longer be a simple gradient, instead, it is a principled negotiation: finding a new set of model parameters $\theta_{k+1}$ that balances proximity to the solution of the new task and proximity to a state that respects old knowledge. In our model, stability and plasticity are interdependent, but not in the oppositional. Unlike prior splitting-based CL (Polson et al., 2015; Yoo et al., 2024; Wang et al., 2025) that still balance penalties, our formulation treats stability as a guide for plasticity, shaping learning rather than simply constraining it.

Our approach yields several key advantages: ① DRS handles the two functions separately via their proximal operators. Instead of mushing them together into a single loss function, the DRS finds a solution point between $f$ and $g$, which can leads to a more stable and negotiation between the

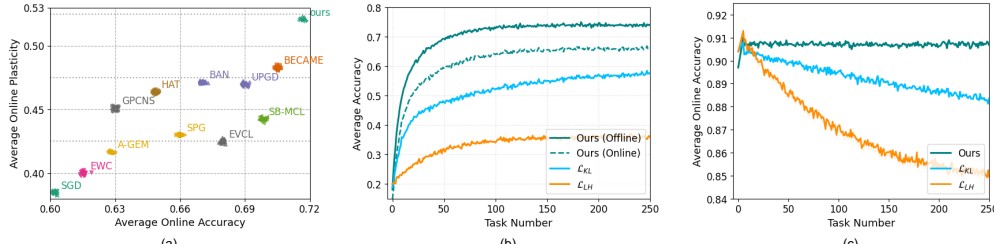

Figure 1: The Stability-Plasticity dilemma in continual learning on EMNIST: (a) Illustrates the trade-off between online average accuracy and plasticity across various methods. Methods closer to the top-right corner better balance the ability to learn new tasks without forgetting. (b) Catastrophic forgetting: average accuracy over seen tasks vs. task index. Forgetful methods drop or remain low; a successful one maintain a consistently high curve throughout training. (c) Loss of plasticity: an ideal learner should maintain a high, stable performance on new tasks regardless of how many it has seen before. A downward-sloping curve on this plot is a sign that the model is losing its plasticity.

two objectives; ② the use of a Bayesian prior provides a structured latent space that facilitates the transfer of shared representations; ③ this is made robust by our use of a flexible Rényi divergence to enforce consistency with the Bayesian prior. Together, these elements create a structured latent space that facilitates knowledge transfer, shifting continual learning from a zero-sum trade-off to a synergistic process. Fig. 1 illustrates two fundamental failures of continual learning: catastrophic forgetting and loss of plasticity. The tasks are designed to be highly coherent, so features learned in one task should accelerate performance on subsequent tasks. However, when trained with a variational inference (Eq. 1), which include KL-divergence, the learner fails to improve across tasks (Fig. 1a), showing repeated forgetting and relearning. A second issue is loss of plasticity, as the model parameters become entrenched to protect old knowledge, its ability to learn new tasks diminishes over time (Lyle et al., 2024; Lee et al., 2023; Bonnet et al., 2025). This is evident in Fig. 1b, where accuracy on new tasks declines with task number. These results highlight the limitations of treating CL as a simple trade-off and motivate the need for our proposed approach.

## 2 RELATED WORK

**Continual learning (CL)** is widely recognized as a foundational requirement for building adaptable and general artificial intelligence systems. A successful CL model must be able to acquire new knowledge while preserving all previously seen tasks. However, standard neural networks, when trained on a new task, tend to overwrite the parameters essential for past tasks, leading to a drastic drop in performance on prior knowledge. This phenomenon is known as catastrophic forgetting (French, 1999), which creates the core stability-plasticity dilemma. Most CL frameworks address this by adding a stability constraint to the new task's loss, forcing the two objectives into a direct and often conflicting summation (Polson et al., 2015). The popular solution families include replay-based methods (Rudner et al., 2022; Hayes et al., 2020; Eskandar et al., 2025), parameter isolation methods (Konishi et al., 2023; Kang et al., 2022; Malviya et al., 2022), and regularization-based methods (Batten et al., 2024; Dohare et al., 2024; Thapa & Li, 2024). Among these, regularization-based methods have gained prominence due to their theoretically motivated approach to managing the stability-plasticity trade-off (Van de Ven et al., 2024). These methods seek to preserve prior knowledge by penalizing updates that would significantly alter parameters important for previously learned tasks. A notable example, EWC (Kirkpatrick et al., 2017), employs the Fisher Information Matrix to identify important weights and imposes a quadratic penalty on their changes. Similarly, VCL (Nguyen et al., 2018) and its extensions (Ahn et al., 2019; Lee & Storkey, 2024; Dhir et al., 2024; Thapa & Li, 2024) adopt a Bayesian perspective, regularizing the model's posterior distribution between tasks to maintain knowledge retention. Further innovations like SFSVI (Rudner et al., 2022) shift from parameter regularization to function space regularization. Despite these advances, a critical limitation persists: the optimization process itself remains forgetful. Most approaches combine task loss and memory regularization into a single objective and optimize it via standard optimizers like SGD, which have no intrinsic mechanism to manage the conflict between competing objectives (Polson et al., 2015; Lee et al., 2023; Wang et al., 2025). As a result, models either

overfit to the new task and forget (too much plasticity) or over-regularize and fail to adapt (too much stability). Our work addresses this gap by building knowledge retention directly into the optimizer. We are aligned with an emerging body of work (Polson et al., 2015; Yoo et al., 2024) that has begun to explore operator splitting methods for CL.

**Operator splitting solvers:** Douglas-Rachford Splitting (DRS) (Douglas & Rachford, 1956; Gabay & Mercier, 1976) is a classic operator splitting method developed for solving optimization problems of the form: $\min_x f(x) + g(x)$, where, $f$ and $g$ are two separate functions that maybe difficult to optimize together, but handling each function individually is easier. DRS reformulates the problem into two distinct subproblems that are solved sequentially using proximal operators. This decompositional ability has made such methods highly popular for large-scale and complex optimization (Stellato et al., 2020; Garstka et al., 2021; Mai et al., 2022; Aljadaany et al., 2019; Tran Dinh et al., 2021; Anshika et al., 2024; Ozaslan & Jovanović, 2025). Given that catastrophic forgetting can be framed as an optimization conflict between task adaptation $f$, and knowledge retention $g$, this splitting provides a principled solution. The DRS algorithm first computes a solution that satisfies the plasticity objective, then refines this solution to be consistent with the stability objective. A proximal operator blends the two solutions, ensuring a balanced update. This deep integration of stability distinguishes our approach from other methods that have explored proximal objectives. For instance, Yoo et al. (2024) use a proximal point objective to stabilize replay-based training, applying a single proximal to the combined task and replay loss. In contrast, our DRS-based continual learner is replay-free and performs a decoupling, splitting the objective into its distinct plasticity and stability components and addressing them in a structured negotiation. In this way, knowledge retention is embedded directly into the optimization process, not added as an external penalty.

**Different from exiting approaches:** First, in contrast to methods like UCL (Ahn et al., 2019), EWC (Kirkpatrick et al., 2017), SB-MCL (Lee et al., 2024) that combine task and regularization losses into a single objective optimized via standard SGD, we reframe the problem as an optimization conflict resolved through Douglas-Rachford Splitting (DRS). This embeds knowledge retention into the optimizer's update rule. Second, our model is entirely replay-free. While coreset-based methods (Borsos et al., 2020; Batra & Clark, 2024; Thapa & Li, 2024) store subsets of past tasks to preserve knowledge, our model operates through the more efficient probabilistic mechanism of posterior propagation, avoiding explicit data storage.

## 3 OUR APPROACH

### 3.1 PROBLEM OVERVIEW

We consider a sequence of tasks $D = \{D^{(1)}, \ldots, D^{(T)}\}$, where each $D^{(t)} = \{(x_n^{(t)}, y_n^{(t)})\}_{n=1}^N$ consists of $N$ input-target pairs. Our goal is to learn these tasks sequentially while preserving and leveraging prior knowledge to achieve synergy, where old knowledge accelerates new learning. For each input $x_n^{(t)}$ from a task $D^{(t)}$, an encoder $(\phi)$ infers a posterior distribution over a shared latent space $z$ by $q_\phi(z \mid x_n^{(t)}) = \mathcal{N}(\mu_\phi(x_n^{(t)}), \operatorname{diag}(\sigma_\phi(x_n^{(t)})^2))$. A shared decoder $(\theta)$, then predicts the output via the likelihood $p_\theta(y_n^{(t)} \mid x_n^{(t)}, z)$. To accumulate knowledge, we adopt a posterior-to-prior propagation (Konishi et al., 2023; Bonnet et al., 2025): after learning task $t-1$, its posterior becomes the prior for task $t$. Specifically, we start with a Gaussian prior, $p(z \mid D^{(0)}) = \mathcal{N}(0, I)$, and for subsequent tasks $(t > 1)$, we set the prior as $p(z \mid D^{(1:t-1)}) = q_{\phi_{t-1}}(z \mid D^{(t-1)})$. As detailed in Appendix A.1, this prior is parameterized as a Gaussian aggregated over the previous dataset, providing a compact summary of acquired knowledge. Then, the training objective for task $t$ is

$$\mathcal{L}(\phi, \theta) = \mathbb{E}_{z \sim q_\phi(z \mid D^{(t)})}\Big[\sum_{n=1}^N \log p_\theta(y_n^{(t)} \mid x_n^{(t)}, z)\Big] - \lambda \sum_{i=1}^d w_i \, D_\alpha(q_\phi^i \parallel p^i), \quad (1)$$

The first term is the maximum likelihood (learning the current task) and the second term is stability (alignment with the prior). The stability is a weighted Rényi divergence (Li & Turner, 2016) between the posterior and prior for each latent dimension $i$. The weights, $w_i = (\sigma_p^i)^2 / \sum_j (\sigma_p^j)^2$, relax the constraints on latent dimensions where the prior is uncertain (high variance) $(\sigma_p^i)$, allowing for plastic adaptation while enforcing stability on learned features. We restrict all distributions to the Gaussian family, for which the Rényi divergence has a closed-form solution (Margossian et al.,

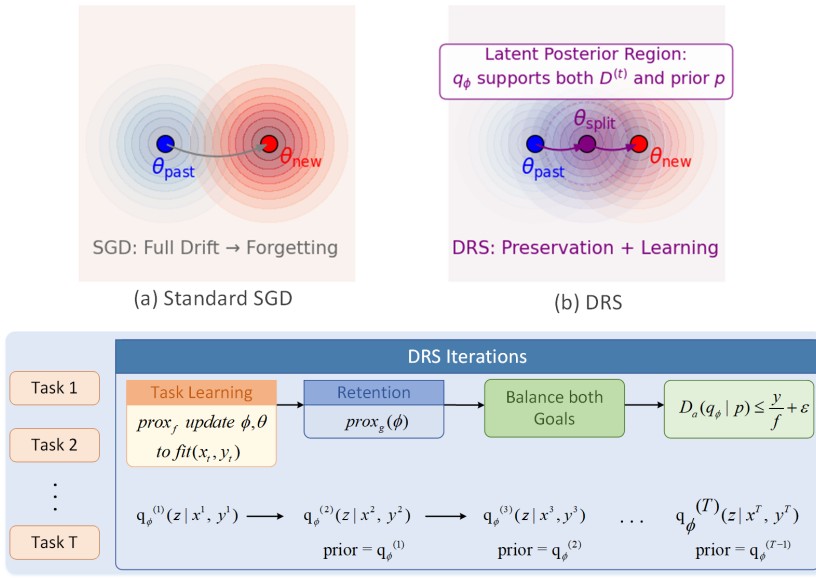

Figure 2: Addressing Catastrophic forgetting with Douglas-Rachford Splitting (DRS). (a) SGD optimizes only for the current/new task, causing the latent posterior $q_\phi$ to drift toward the new distribution, leading to forgetting of past knowledge ($\theta_{\text{past}}$). (b) DRS constrains the posterior within a region that supports both old and new task distributions, preserving prior knowledge. $\theta_{\text{past}}, \theta_{\text{new}}, \theta_{\text{split}}$ represents the past, new and the balanced posteriors. (c) Our optimization loop: task-specific learning, retention, and a relaxation step that balances both forces. This structure avoids gradient interference and supports continual learning across long task sequences. Our Algorithm is in (3.1.1).

2024) (see Appendix A.4.1). Most continual learning methods rely on KL-divergence (Dhir et al., 2024; Bonnet et al., 2025; Eskandar et al., 2025), in Appendix A.4 we argue that Rényi divergence provides a more flexible and effective constraint. However, our contribution is an optimization scheme based on Douglas-Rachford Splitting (DRS) that decouples the plasticity and stability terms into proximal subproblems, enabling synergistic learning of new tasks while preserving prior knowledge.

### 3.1.1 DRS-BASED CONTINUAL LEARNER.

To optimize Eq. 1, we reformulate the problem to leverage the power of operator splitting, where

- $f(\phi, \theta) = -\mathbb{E}_{z \sim q_\phi}[\sum \log p_\theta(y_n \mid x_n, z)]$, (task-fitting / plasticity),

- $g(\phi) = \lambda \sum_{i=1}^{d} w_i \, \mathrm{D}_\alpha \left( q_\phi^i \parallel p^i \right)$, (prior-alignment / stability).

The term $f$ depends on both $\phi$ (via $q_\phi$) and $\theta$ (via $p_\theta$), while the stability $g$ only depends on the encoder $\phi$. This structure makes the problem ideally suited for DRS, which handles the two terms in separate proximal steps. The algorithm iterates over an auxiliary variable $u_i = (\phi_i, \theta_i)$, initialized for task $t$ as $u_0 = (\phi_{t-1}, \theta_{t-1})$. For iterations $i = 1, \ldots, I$, we perform the following steps

**1. Task-Fitting Proximal (Plasticity):** First, we compute the proximal operator for the plasticity objective $f$, which updates the model parameters to learn the current task

$$x_i = \text{prox}_f(u_{i-1}) = \arg \min_{\phi, \theta}[f(\phi, \theta) + \frac{1}{2\gamma}\|(\phi, \theta) - u_{i-1}\|^2]. \tag{2}$$

This step updates both $\phi$ and $\theta$. As this problem is nonconvex (Aljadaany et al., 2019), we approximate the solution via gradient-based updates (using Adam) initialized from $u_{i-1}$.

**2. Prior-Alignment Reflection (Stability):** Next, we compute the proximal operator for $g$, applied to a reflection of the plasticity output

$$y_i = \text{prox}_g(2x_i - u_{i-1}) = \arg\min_\phi [g(\phi) + \frac{1}{2\gamma}\|\phi - (2x_i^\phi - u_{i-1}^\phi)\|^2]. \tag{3}$$

This step updates only the encoder $\phi$ to align its posterior with the prior, as detailed in Appendix A.4.1. The decoder parameters $\theta$ are passed unchanged from the previous step ($y_i = (y_i^\phi, x_i^\theta)$), preserving their specialization on the new task. The encoder ($\phi$) thus mediates between task fitting and prior alignment, as $\phi$ defines both $q_\phi(z)$ and the divergence constraint. As we prove in Proposition 3.1, the Rényi divergence is essential for a robust, DRS-based continual learner.

**3. Relaxed Update:** Finally, we update the auxiliary variable by moving towards the refined state

$$u_i = u_{i-1} + \lambda_r(y_i - x_i). \tag{4}$$

This step interpolates between the plasticity $x_i$ and the stability refinement $y_i$. After $I$ iterations, the final parameters for task $t$ are set to $(\phi_t, \theta_t) = x_I$, and knowledge is propagated by setting the next prior as $p(z \mid D^{(1:t)}) = q_{\phi_t}(z \mid D^{(t)})$. Our model is detailed in Algorithm 3.1.1 and notation details in Appendix A.2. As we prove in Proposition 3.2, this DRS-based optimization is guaranteed to converge to a stationary point of the continual learning objective. Our model finds a principled compromise between the competing goals of plasticity and stability. Specifically, stationary points imply that both objectives $(f, g)$ are simultaneously satisfied, and the vanishing discrepancy between iterations shows that stability complements plasticity rather than conflicting with it.

**Proposition 3.1.** *Let posterior $q(z) = \mathcal{N}(z|\mu_q, \Sigma_q)$ and prior $p(z) = \mathcal{N}(z|\mu_q, \Sigma_p)$ be Gaussian distributions. Consider the proximal operator problem, $q^\star = \arg\min_q[D(q \parallel p) + \frac{1}{2\gamma}D(q \parallel v)]$, where $v(z) = \mathcal{N}(z|\mu_v, \Sigma_v)$ is the plastic proposal. When $\mu_v$ lies outside the high-probability region of $p$, the KL-divergence $(D_{KL})$ becomes dominated by the stability term $g$, while the Rényi divergence, $prox_{g_{RD}}$ maintains balance between plasticity and stability.*

*Proof.* The proximal update solves the optimization problem $\min_q[D(q \parallel p) + \frac{1}{2\gamma}\|\mu_q - \mu_v\|^2]$, (Galke et al., 2024). The KL divergence possesses a zero-forcing (Li & Turner, 2016) behavior, meaning $D_k(g \parallel p) \to \infty$ if $g$ places mass where $p(z) = 0$. When the plastic $x_i$ proposes parameters $\mu_v$ far from the prior support $p$, any $g^\star$ with mean near $\mu_v$ will have significant mass, where $p(z) \approx 0$. The zero-forcing property forces the optimizer to ignore $v$ and collapse $g^\star$ onto the prior's support to avoid infinite penalty (Margossian et al., 2024). In contrast, the Rényi divergence $D_\alpha(q \parallel p) = \frac{1}{\alpha-1}\log\int p(z)^\alpha q(z)^{1-\alpha}dz$ is zero-avoiding (Bresch & Stein, 2024; Galke et al., 2024). We consider the same scenario, where the proposal $v$ is far from the prior $p$. The term $p(z)^\alpha q(z)^{1-\alpha}$ remains bounded when $p(z) \approx 0$, so the penalty is finite. The optimizer can thus find a compromise posterior $q^\star$ near the proposal $v$, while paying a reasonable penalty for disagreeing with the prior; allowing meaningful interpolation between $\mu_p$ and $\mu_v$ (see Appendix A.2).

**Proposition 3.2.** *Let $F(\omega) = f(\omega) + g(\omega)$ be the continual learning objective, where $f$ is the task-learning term (plasticity), and $g$ is the prior-alignment term (stability). Consider the DRS iterations from Eqs. 2, 3 and 4*

$$x_k = prox_f(u_k), \qquad y_k = prox_g(2x_k - u_k), \qquad u_{k+1} = u_k + \lambda_r(y_k - x_k),$$

*then the following hold: (i) any fixed point of the DRS corresponds to a stationary point $\omega^\star$, satisfying the first-order optimality condition $0 \in \nabla f(\omega^\star) + \partial g(\omega^\star)$; (ii) the iterates converge in the sense that the discrepancy between the plasticity and stability steps vanishes, i.e., $\lim_{k\to\infty}\|x_k - y_k\| = 0$.*

*Proof.* When our algorithm has found its optimal solution (fixed point $u^\star$), and stops changing, a consequence of the update is that the plasticity $(x^\star)$ and stability $(y^\star)$ must have become identical $y^\star = x^\star \triangleq w^\star$, (see Appendix A.3.2). From the optimality conditions of the two proximal steps

- $x^\star = \text{prox}_f(u^\star) \Rightarrow u^\star = w^\star + \gamma\nabla f(w^\star)$,
- $y^\star = \text{prox}_g(2x^\star - u^\star) \Rightarrow \frac{(2w^\star - u^\star) - w^\star}{\gamma} \in \partial g(w^\star)$.

---

**Algorithm 1** Optimizing Continual Learning via Douglas-Rachford Splitting

---

**Require:** Sequence of datasets $D^{(T)}$, iterations $I$, inner loop $K$ for $\text{prox}_f$, $\gamma$, $\lambda$.

1: Initialize $\phi^{(0)}, \theta^{(0)}$; prior $p(z) \leftarrow N(0, I)$.
2: **for** task $t = 1$ to $T$ **do**
3:     Receive data $D_t = \{(x_n, y_n)\}_{n=1}^N$.
4:     Set variable $u_0 \leftarrow (\phi_{t-1}, \theta_{t-1})$.
5:     **for** iteration $i = 1$ to $I$ **do**
6:         **Step 1: Plasticity (Task-Fitting Proximal Step)**
7:             $x_i \leftarrow \text{prox}_f(u_{i-1})$.
8:         // Approximated with $K$ gradient steps, initialized from $u_{i-1}$.
9:         **Step 2: Stability (Prior-Alignment Reflection Step)**
10:         $y_i \leftarrow \text{prox}_g(2x_i - u_{i-1})$.
11:         // Only updates encoder $\phi$. Decoder is passed through $y_i = (y_i^\phi, x_i^\theta)$.
12:         **Step 3: Relaxed Update**
13:         $u_i \leftarrow u_{i-1} + \lambda(y_i - x_i)$.
14:     **end for**
15:     // Update model and prior for the next task
16:     Set final parameters for task $t$: $(\phi_t, \theta_t) \leftarrow x_I$.
17:     Update prior for task $t + 1$: $p(z) \leftarrow q_{\phi_t}(z \mid D^{(t)})$.
18: **end for**
19: **return** final model parameter $\{\phi_T, \theta_T\}$.

---

Substituting the first into the second gives $\frac{w^\star - (w^\star + \gamma \nabla f(w^\star))}{\gamma} \in \partial_g(w^\star)$, where yields $-\nabla f(w^\star) \in \partial_g(w^\star) \Rightarrow 0 \in f(w^\star) + \partial_g(w^\star)$. This is the first-order stationarity condition (Polson et al., 2015; Aragón Artacho et al., 2020; Ozaslan & Jovanović, 2025) for the composite objective $F$, implying that the adjustment to parameters is satisfied by stability and plasticity, ensuring coordination rather than conflict. Additionally, the DRS update is a firmly non-expansive (Eckstein & Bertsekas, 1992; Aljadaany et al., 2019), where the sequence of iterates is monotone with respect to the fixed points. This provides the inequality: $\|u_{k+1} - u^\star\|^2 \leq \|u_k - u^\star\|^2 - \lambda_r(2 - \lambda_r)\|x_k - y_k\|^2$. Since $\|u_k - u^\star\|^2$ is non-increasing, so it converges monotonically (Anshika et al., 2024; Aragón Artacho et al., 2020). As the sequence converges, the difference between consecutive terms must approach zero $\lim(\|u_{k+1} - u^\star\|^2 - \|u_k - u^\star\|^2) = 0$. For the inequality above to hold, the final term must also vanish. Since $\lambda_r(2 - \lambda_r) > 0$, we have $\lim_{k \to \infty} \|x_k - y_k\| = 0$ (more detail in Appendix A.3).

**Discussion.** Our theoretical analysis justifies the proposed learning strategy. The DRS-based continual learner is guaranteed to converge to a stationary point (a principled compromise between plasticity and stability), evidenced by the vanishing discrepancy between the two steps ($\|x_k - y_k\| \to 0$). More importantly, our analysis reveals that the robust negotiation is possible because of using Rényi divergence. We proved that it remains well-posed when learning novel tasks, a scenario where the standard KL divergence may fail (Galke et al., 2024; Bresch & Stein, 2024). Indeed, DRS has the strongest theoretical guarantees in convex (Eckstein & Bertsekas, 1992; Garstka et al., 2021; Mai et al., 2022) and nonconvex (Polson et al., 2015; Li & Pong, 2016; Aragón Artacho et al., 2020; Tran Dinh et al., 2021) settings. By reformulating the problem at the optimization level, we create a more effective continual learner that avoids the zero-sum trade-offs of prior methods. Additional analyses, including computational complexity, are provided in Appendices A.3, A.4 and A.3.3.

## 4 EXPERIMENTS

We evaluate our model on EMNIST (Cohen et al., 2017), CIFAR-10/100 (Krizhevsky et al., 2009), ImageNet (Deng et al., 2009), TinyImageNet (Wu et al., 2017) and Celeb (Guo et al., 2016) datasets (details in Appendix B.1), with learners that use multi-layer perceptrons, convolutional neural networks (see Appendix B.2), and residual neural networks (He et al., 2016). For other baselines, we used their codes and replacing their backbones to any of these for fair comparison. Our model is compared to suitable baselines: EWC (Kirkpatrick et al., 2017), IBPCL (Kumar et al., 2021), A-GEM (Chaudhry et al., 2019), SB-MCL (Lee et al., 2024), UCL (Ahn et al., 2019), TAG (Malviya et al., 2022), EVCL (Batra & Clark, 2024), UPGD (Elsayed & Mahmood, 2024), POCL (Wu et al., 2024), HAT (Serra et al., 2018), BAN (Thapa & Li, 2024), SPG (Konishi et al., 2023) and WSN

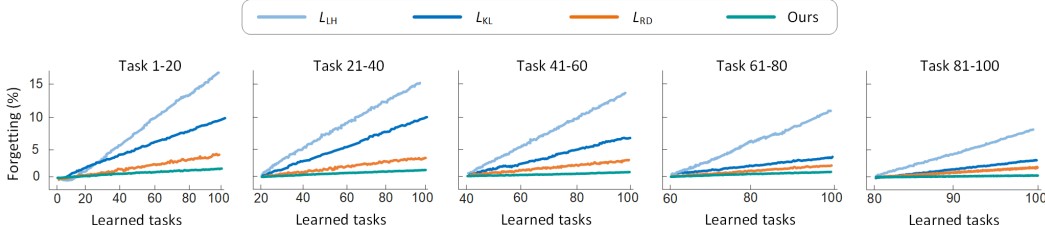

Figure 3: Forgetting analysis over 100 tasks on the CASIA classification benchmark. Each sub-plot summarizes average forgetting across 20-task intervals, and the final plot shows the average forgetting across all 100 tasks. We compare $\mathcal{L}_{LH}$ (likelihood only; no stability term), $\mathcal{L}_{KL}$ (KL regularizer), $\mathcal{L}_{RD}$ (Rényi divergence, but standard gradient updates), and Ours (DRS + Rényi).

(Kang et al., 2022). The hyperparameters are set to $\gamma = 0.5$, and $\lambda = 0.7$. *Here, we focus on the key results and provide additional results in the Appendix C.* Following the baselines, we evaluate the performance using three metrics. Average Accuracy (ACC): The mean classification accuracy across all tasks computed after training on each task. Backward Transfer (BT): The change in accuracy on previous tasks after training on a new task, measuring forgetting (positive values indicate improvement, negative indicate forgetting). Forward Transfer (FT): The improvement in accuracy on a new task due to knowledge from previous tasks, assessing loss of plasticity.

## 4.1 Results Against Catastrophic Forgetting and Loss of Plasticity

We evaluate our model on six benchmarks: CIFAR-100 split into 10 tasks (10 classes per task) and 20 tasks (5 classes per task), Tiny-ImageNet with 20 tasks (5 classes per task), ImageNet with 100 tasks (10 classes per task), CelebA with 10 tasks (celebrity identities), and EMNIST with 10 tasks (handwritten symbols). For CIFAR-100, Tiny-ImageNet, and ImageNet, classes are disjoint across tasks (each task has a distinct set of classes), where catastrophic forgetting is the primary challenge. In contrast, CelebA and EMNIST share the same label space across tasks (joint tasks), where forward/backward transfer is more critical than retention. All methods use ResNet-18 as the backbone. Accuracy results are reported in Table 1. On the four disjoint-task benchmarks, our model achieves the best average accuracy (**65.7%**), while demonstrating near-minimal forgetting with average BWT **-1.9** (in Table 2). In the joint-task setting, our approach achieves the highest accuracy (**88.2%**) and the largest positive backward transfer (BWT **+3.2**), indicating that DRS + Rényi reaches a superior stability-plasticity trade-off. The model also demonstrates strong forward transfer in Table 3, accelerating new task learning by up to **+10.4**. These metrics confirm that our approach achieves high accuracy, low forgetting, and strong forward transfer, setting a new standard for continual learning.

## 4.2 Forgetting Analysis

Figure 3 analyzes forgetting behaviour on CASIA-100 (Liu et al., 2011), over 100 sequential tasks. Following the metric from (Chaudhry et al., 2018), we measure forgetting for a task $t$ as the drop in its accuracy after the model has trained on subsequent tasks $t' > t$. To visualize this long sequence, the figure is split into five subplots, each showing the average forgetting over 20-task intervals; the final plot summarizes the average forgetting across all 100 tasks. A small CNN with ReLU activations is used (Appendix B.2). We compare our full DRS-based model against three baselines trained with standard gradient-based updates (Eq. 1). Likelihood-Only ($\mathcal{L}_{LH}$), where $\alpha = 0$: The model is trained only on the task-fitting term of Eq. (1), with no stability constraint. KL-based ($\mathcal{L}_{KL}$): The model is trained on the full objective but with the standard KL divergence. Rényi-based ($\mathcal{L}_{RD}$): The model is trained on the full objective with Rényi divergence, but without our proposed DRS optimization (using the training objective in Eq. 1). The results demonstrate the superiority of our proposed model. Using $\mathcal{L}_{LH}$ shows catastrophic forgetting, as expected. For the earliest tasks (1-20), its forgetting rate climbs sharply, reaching over 15% by the end of the sequence. Adding KL divergence improves upon LH but still accumulates significant forgetting, exceeding 13% for the first block of tasks. In contrast, our model demonstrates the least forgetting (close to zero), and remaining

| Method | Disjoint tasks | | | | | Joint tasks | | |
|---|---|---|---|---|---|---|---|---|
| | C100 [10] | C100 [20] | TIN [20] | IN [100] | Avg. | CelebA [10] | EM [10] | Avg. |
| Mix | $75.1 \pm 0.3$ | $79.8 \pm 0.4$ | $52.1 \pm 0.3$ | $62.7 \pm 0.4$ | 67.4 | $87.9 \pm 0.7$ | $86.3 \pm 0.7$ | 87.1 |
| Single | $67.9 \pm 2.1$ | $77.0 \pm 0.4$ | $43.8 \pm 2.6$ | $46.3 \pm 0.4$ | 58.9 | $76.5 \pm 1.9$ | $81.3 \pm 0.9$ | 78.9 |
| A-GEM | $51.4 \pm 1.2$ | $56.9 \pm 5.3$ | $37.5 \pm 0.6$ | $34.2 \pm 0.9$ | 45.0 | $84.6 \pm 2.1$ | $86.9 \pm 0.2$ | 85.8 |
| EWC | $61.7 \pm 1.0$ | $65.1 \pm 2.3$ | $41.5 \pm 0.9$ | $28.2 \pm 1.2$ | 49.2 | $81.9 \pm 2.4$ | $86.8 \pm 0.5$ | 84.3 |
| BAN | $71.6 \pm 0.5$ | $78.4 \pm 0.4$ | $50.6 \pm 0.4$ | $57.6 \pm 0.5$ | 64.4 | $\underline{87.2 \pm 0.7}$ | $87.6 \pm 0.2$ | 87.4 |
| SB-MCL | $\mathbf{72.3 \pm 0.3}$ | $78.1 \pm 0.2$ | $50.8 \pm 0.7$ | $\underline{58.6 \pm 0.8}$ | $\underline{64.9}$ | $86.9 \pm 0.9$ | $\underline{88.1 \pm 0.3}$ | $\underline{87.5}$ |
| HAT | $71.2 \pm 0.4$ | $75.2 \pm 0.5$ | $45.8 \pm 1.8$ | $45.9 \pm 1.5$ | 59.6 | $79.6 \pm 2.3$ | $84.9 \pm 0.8$ | 82.3 |
| SPG | $69.2 \pm 0.3$ | $76.5 \pm 0.8$ | $49.7 \pm 0.2$ | $\underline{58.6 \pm 0.5}$ | 63.5 | $87.1 \pm 0.9$ | $87.9 \pm 0.2$ | 87.5 |
| IBPCL | $68.7 \pm 1.0$ | $77.3 \pm 0.9$ | $48.6 \pm 0.6$ | $55.2 \pm 0.7$ | 62.5 | $85.2 \pm 0.5$ | $86.5 \pm 0.4$ | 85.9 |
| UCL | $64.9 \pm 0.8$ | $73.6 \pm 0.6$ | $46.5 \pm 0.6$ | $39.1 \pm 0.7$ | 56.1 | $86.4 \pm 0.5$ | $85.7 \pm 1.2$ | 86.0 |
| UPGD | $71.4 \pm 0.2$ | $77.5 \pm 0.5$ | $\underline{51.2 \pm 0.3}$ | $58.0 \pm 0.4$ | 64.5 | $85.9 \pm 0.4$ | $87.5 \pm 0.3$ | 86.7 |
| POCL | $70.2 \pm 0.6$ | $\underline{79.0 \pm 1.2}$ | $49.8 \pm 0.6$ | $57.2 \pm 0.7$ | 64.1 | $85.2 \pm 0.9$ | $87.1 \pm 0.6$ | 86.2 |
| TAG | $61.0 \pm 0.5$ | $68.7 \pm 0.9$ | $43.5 \pm 0.7$ | $45.8 \pm 0.2$ | 54.8 | $76.3 \pm 1.9$ | $84.5 \pm 0.5$ | 80.4 |
| WSN | $70.4 \pm 0.2$ | $77.5 \pm 0.5$ | $47.9 \pm 0.4$ | $52.1 \pm 0.4$ | 62.1 | $84.2 \pm 1.1$ | $86.7 \pm 0.3$ | 85.5 |
| Ours | $71.8 \pm 0.3$ | $\mathbf{79.5 \pm 0.6}$ | $\mathbf{51.6 \pm 0.4}$ | $\mathbf{59.7 \pm 0.5}$ | $\mathbf{65.7}$ | $\mathbf{87.9 \pm 0.5}$ | $\mathbf{88.6 \pm 0.1}$ | $\mathbf{88.2}$ |

Table 1: Accuracy (%) results for joint and disjoint task settings on CIFAR-100 (C100 [10], C100 [20]), Tiny-ImageNet (TIN [20]), ImageNet (IN [100]), MS-Celeb (CelebA [10]), and EMNIST (EM [10]) datasets. 'Mix' refers to training all tasks together, and 'Single' refers to learning a separate model for each task. The results show that our model outperforms other methods across most settings. **Bold** and underlined text represents the best and the second-best results, respectively.

| Method | Disjoint tasks | | | | | Joint tasks | | |
|---|---|---|---|---|---|---|---|---|
| | C100 [10] | C100 [20] | TIN [20] | IN [100] | Avg. | CelebA [10] | EM [10] | Avg. |
| A-GEM | -12.4 | -19.5 | -8.5 | -14.7 | -13.8 | +0.5 | +1.3 | +0.9 |
| EWC | -6.1 | -11.8 | -6.9 | -21.3 | -11.5 | -0.9 | +1.2 | +0.1 |
| BAN | -3.2 | -4.7 | -3.6 | -2.5 | -3.5 | +2.1 | +1.4 | +1.7 |
| SB-MCL | -2.5 | -3.9 | -3.4 | -2.3 | -3.0 | +2.9 | +1.2 | +2.0 |
| SPG | -4.7 | -5.1 | -3.9 | -1.7 | -3.9 | +2.7 | +0.8 | +1.8 |
| IBPCL | -4.6 | -6.1 | -5.2 | -4.6 | -5.1 | +1.6 | +0.9 | +1.3 |
| UCL | -5.9 | -8.5 | -7.6 | -13.9 | -9.0 | $\mathbf{+3.2}$ | +0.8 | $\underline{+2.0}$ |
| UPGD | -2.6 | -3.2 | -2.8 | -2.5 | -2.8 | +2.4 | +1.5 | +2.0 |
| POCL | -2.3 | -2.9 | -3.4 | -3.5 | -3.0 | +2.1 | $\underline{+1.7}$ | +1.9 |
| TAG | $\mathbf{-0.9}$ | $\mathbf{-1.8}$ | $\underline{-1.3}$ | $\mathbf{-0.7}$ | $\mathbf{-1.2}$ | +0.5 | -0.1 | +0.2 |
| Ours | $\underline{-1.6}$ | $\underline{-2.3}$ | $\underline{-2.5}$ | $\underline{-1.3}$ | $\underline{-1.9}$ | $\mathbf{+3.9}$ | $\mathbf{+2.4}$ | $\mathbf{+3.2}$ |

Table 2: Backward transfer (BWT) result. Negative values indicate forgetting (degradation), while positive values indicate improvement due to knowledge transfer. Our method achieves the best average knowledge transfer (+ 3.2) on joint tasks and minimized forgetting (- 1.9) on disjoint tasks. **Bold** and underlined text represents the best and the second-best results, respectively.

below 4% across all intervals. Even without DRS, Rényi outperforms KL (we demonstrate this in Appendix A.4 and A.4.1), but the full combination is consistently best.

### 4.3  ABLATION STUDY

For ablation studies, we follow a simplified setting and use a two-layer MLP (1024-512, ReLU). Each model is trained on CIFAR100 (20 tasks) using a total of 10,000 update steps. Results are averaged over 5 seeds, and we report relative computation time and accuracy in Fig. 4. We focus on two core components of our model: (i) stochastic Gaussian latent encoding, and (ii) divergence parameter $\alpha$. **Effect of latent stochasticity:** Our model samples $z \sim \mathcal{N}(\mu_\phi, \sigma_\phi^2)$; the ablation uses a deterministic latent $z' = \mu_\phi$ (i.e., no sampling in the forward pass $\sigma_\phi = 0$) [1]. Panel (b) shows this reduces training time by about 9%, but it also leads to a performance drop, reducing average

---
[1] We still predict $\sigma_\phi$ and use it in the stability term; only sampling is removed.

| Method | Disjoint tasks | | | | | Joint tasks | | |
|---|---|---|---|---|---|---|---|---|
| | C-100 [10] | C-100 [20] | MIN [20] | IN [100] | Avg. | CelebA [10] | EM [10] | Avg. |
| A-GEM | -2.7 | -0.8 | -3.5 | -0.6 | -1.9 | +8.1 | +4.5 | +6.3 |
| EWC | +0.6 | -1.9 | -4.3 | -1.1 | -1.7 | +7.3 | +4.9 | +6.1 |
| BAN | +6.2 | +4.8 | +6.8 | +9.2 | +6.7 | +9.8 | +7.4 | +8.6 |
| SB-MCL | **+7.1** | +4.3 | +8.1 | +8.7 | +7.1 | +10.5 | +6.9 | +8.7 |
| SPG | +5.7 | +4.7 | +7.6 | +9.5 | +6.9 | +9.4 | +5.7 | +7.6 |
| IBPCL | +3.8 | +3.1 | +5.4 | +6.3 | +4.7 | +8.6 | +4.9 | +6.8 |
| UCL | +4.1 | **+6.1** | +7.8 | +3.9 | +5.5 | +8.5 | +3.7 | +6.1 |
| UPGD | +5.6 | +4.5 | +7.6 | +9.3 | +6.8 | +10.2 | +7.8 | +9.0 |
| POCL | +5.4 | +5.2 | +7.1 | +7.9 | +6.4 | +10.8 | +7.3 | +9.1 |
| TAG | -4.3 | -5.1 | -4.6 | -4.1 | -4.5 | -0.2 | +3.7 | +1.8 |
| **Ours** | +6.5 | +5.9 | **+8.4** | **+10.7** | **+7.9** | **+12.3** | **+8.5** | **+10.4** |

Table 3: Forward transfer (FT) results on disjoint and joint tasks. Our model achieves the highest average performance, outperforming the next best method (UPGD) by 10% and 16% respectively. **Bold** and underlined text represents the best and the second-best results.

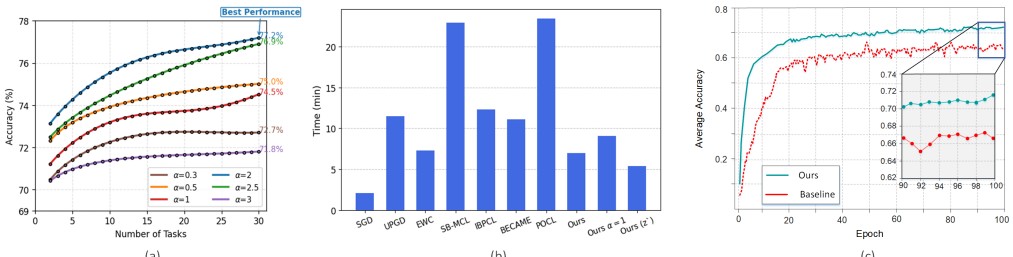

(a)  (b)  (c)

Figure 4: (a) Ablation study on CIFAR100 (20 tasks) benchmark. **(a)** Average accuracy across different values of the divergence parameter $\alpha$. The best result is achieved at $\alpha = 2.0$, reaching an average accuracy of $\approx 77\%$, while the lowest performance occurs at $\alpha = 0.0$, dropping to $\approx 72\%$. **(b)** Relative training time for various methods using a RTX-3090 GPU. Our DRS-based continual learner achieves competitive runtime while outperforming all baselines in accuracy. SGD corresponds to standard optimization without DRS (i.e., direct minimization of Eq. 1). The variant without latent sampling ($z'$) reduces compute time by 9%, but results in lower final accuracy. **(c)** Performance of KL-divergence (baselines) vs. D-divergence (our model). Using KL ($\alpha = 1$) degrades the performance, and our model ($\alpha = 2$) consistently achieves higher accuracy and stability.

accuracy from 79.1% to 76.3%, highlighting the importance of uncertainty modeling in continual learning. **Effect of Rényi:** We vary $\alpha \in \{0.3, 0.5, 1.0, 2.0, 2.5\}$. Performance is lowest for small $\alpha$ that are too permissive of forgetting, while the KL-divergence equivalent ($\alpha = 1.0$ red line) is also suboptimal. The best performance is achieved in the range of $\alpha \in [2.0, 2.5]$, i.e, as predicted by the prox weighting $\frac{\lambda w \alpha \gamma}{\alpha \sigma_p^2 + (1-\alpha)\sigma_q^2}$, which increases with $\alpha > 1$, strengthening adaptive alignment (less drift/forgetting). Panel (b) also shows our method is competitive or faster than many baselines.

## 5 CONCLUSION

Continual learning has long been framed as a trade-off between stability and plasticity, where progress in one dimension comes at the expense of the other. In this paper, we challenged that framing and showed that the true barrier lies in objective entanglement (gradients from new data interfere with useful representations from prior tasks). To address this, we introduced a DRS-based optimization strategy that decouples stability and plasticity via separate proximal operators. This formulation reframes continual learning not as a zero-sum struggle, but as a synergistic process, where prior knowledge guides and accelerates the acquisition of new knowledge. Across multiple benchmarks, our method demonstrates superior performance in terms of stability, adaptability, and computational efficiency when compared to state-of-the-art baselines.

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

## A  APPENDIX

### A.1  POSTERIOR AND PRIOR CONSTRUCTION

For each input $x_n^{(t)}$ from task $t$, the encoder $\phi$ outputs a Gaussian distribution over the latent variable $z$ as, $q_\phi(z \mid x_n^{(t)}) = \mathcal{N}(\mu_\phi(x_n^{(t)}), \mathrm{diag}(\sigma_\phi(x_n^{(t)})^2))$.. At the dataset level, we approximate the posterior as a mean-field product across examples $q_\phi(z \mid D^{(t)}) \propto \prod_{n=1}^N q_\phi(z \mid x_n^{(t)})$.. This factorization is used in variational autoencoders methods (Dhir et al., 2024), where the encoder acts as a shared network producing local posterior factors. For the prior, at the start of training, the prior is chosen as an isotropic Gaussian, $p(z \mid D^{(0)}) = \mathcal{N}(0, I)$. For each subsequent task $(t > 1)$, we adopt a Markovian update rule $p(z \mid D^{(1:t-1)}) \approx q_\phi(z \mid D^{(t-1)})$, i.e., the posterior of the previous task serves as the prior for the current one. This compact approximation avoids the need to store all past data, while carrying forward a summary of accumulated knowledge. An alternative approach, explored in Bayesian CL (Dhir et al., 2024; Lee et al., 2024), relies on exponential-family posteriors and conjugate priors. By the Fisher-Darmois-Koopman-Pitman theorem (Koopman, 1936), such families admit exact posterior updates with sufficient statistics that do not grow with the dataset size. Our model differs with these strategies since we do not require conjugacy and instead optimize Gaussian posteriors via DRS. This makes our approach applicable to non-exponential families and complex neural encoders, at the cost of approximate (gradient-based) updates.

### A.2  HYPERPARAMENTS

Table 4: Summary of notation.

| Symbol | Description |
|---|---|
| $D^{(t)} = \{(x_n^{(t)}, y_n^{(t)})\}_{n=1}^N$ | Dataset of task $t$ with $N$ samples |
| $D = \{D^{(1)}, \ldots, D^{(T)}\}$ | Sequence of $T$ tasks |
| $x_n^{(t)} \in \mathbb{R}^m$ | Input of sample $n$ from task $t$ |
| $y_n^{(t)} \in \mathbb{R}^k$ | Target of sample $n$ from task $t$ |
| $z \in \mathbb{R}^d$ | Latent variable (shared space across tasks) |
| $\phi$ | Parameters of encoder network |
| $\theta$ | Parameters of decoder network |
| $q_\phi(z \mid x)$ | Encoder posterior, Gaussian with mean $\mu_\phi(x)$ and variance $\sigma_\phi(x)^2$ |
| $p_\theta(y \mid x, z)$ | Decoder likelihood of target given input and latent |
| $p(z \mid D^{(t)})$ | Task-specific prior over latents (propagated from previous posterior) |
| $f(\phi, \theta)$ | Plasticity term: negative log-likelihood of current task |
| $g(\phi)$ | Stability term: weighted divergence between posterior and prior |
| $\lambda$ | Weighting coefficient for stability term |
| $D_\alpha(\cdot \parallel \cdot)$ | Rényi divergence between two Gaussians |
| $\gamma$ | Proximal regularization parameter |
| $\rho$ | Relaxation parameter in DRS update |
| $u_i = (\phi_i, \theta_i)$ | Auxiliary iterate in DRS optimization |
| $x_i = (\phi_i^x, \theta_i^x)$ | Proximal solution from task-fitting step |
| $y_i = (\phi_i^y, \theta_i^y)$ | Proximal solution from prior-alignment step |

### A.3 THEORETICAL ANALYSIS

We analyze our DRS-based optimization scheme from three perspectives: (1) convergence theory under DRS objectives, and (2) continual learning-specific stability-plasticity trade-offs. Together, these results show that our method not only converges to a stationary point but also progressively aligns stability and plasticity, reducing interference. (3) We also analyze its computational complexity relative to common CL baselines.

#### A.3.1 PLASTICITY-STABILITY CONVERGENCE

Let the model parameters be $\omega = (\phi, \theta)$, and define the composite objective

$$F(\omega) = f(\omega) + g(\omega),$$

where $f$ is the nonconvex task-fitting term (plasticity) and $g$ is the convex prior-alignment term (stability). Each task $t$ requires solving $\min_\omega F(\omega)$ using DRS iterations:

$$x_k = \text{prox}_{\gamma f}(u_k), \tag{5}$$

$$y_k = \text{prox}_{\gamma g}(2x_k - u_k), \tag{6}$$

$$u_{k+1} = u_k + \lambda_r(y_k - x_k). \tag{7}$$

**Assumptions.**

1. (*$L$-smoothness of $f$.*) The task-fitting function $f$ is differentiable with Lipschitz continuous gradient for $L_f > 0$. That is, for any $\omega_1, \omega_2$ as: $\|\nabla f(\omega_1) - \nabla f(\omega_2)\| \leq L_f \|\omega_1 - \omega_2\|$.

2. (**Convexity of $g$.**) The prior-alignment function $g$ is convex, and lower semi-continuous. We also assume its proximal operator, $\text{prox}_{\gamma g}(\cdot)$, can be computed efficiently (e.g., in closed-form with $\alpha = 0.5$).

3. (**Coercivity.**) The overall objective function $F(\omega)$ is coercive, i.e., $F(\omega) \to \infty$ as $\|\omega\| \to \infty$, ensuring the iterates of our algorithm remain in a bounded set.

However, for nonconvex problems like ours, the goal is to prove convergence to a stationary point (Polson et al., 2015; Eckstein & Bertsekas, 1992). A point $\omega^*$ is a stationary point of $F = f + g$ if it satisfies the first-order optimality condition (Li & Pong, 2016): $0 \in \nabla f(\omega^*) + \partial g(\omega^*)$, where $\partial g$ is the subdifferential of $g$.

**Proposition A.1** (Convergence to a stationary point). *Let Assumptions 1–3 hold. Let $\{u_k\}$ be the sequence generated by DRS. Then:*

1. *The sequence $\{u_k\}$ remains bounded.*

2. *The objective decreases monotonically: there exists $C > 0$ such that*

$$F(x_{k+1}) \leq F(x_k) - C\|x_{k+1} - x_k\|^2.$$

3. *Consequently, $\sum_{k=0}^{\infty} \|x_{k+1} - x_k\|^2 < \infty$, which implies $\|x_{k+1} - x_k\| \to 0$.*

4. *Any limit point of $\{x_k\}$ is a stationary point of $F$, i.e. $0 \in \nabla f(\omega^\star) + \partial g(\omega^\star)$.*

*Proof.* The proof follows standard DRS analysis using the Douglas-Rachford envelope (Ozaslan & Jovanović, 2025; Polson et al., 2015). Assumption 1 ensures controlled descent of $f$, while Assumption 2 ensures the stability prox is well-defined. For each iteration $k$, the output $x_{k+1}$ is better than $x_k$. It can be shown from the properties of proximal operators and L-smoothness that there exists a constant $C > 0$ such that, $F(x_{k+1}) \leq F(x_k) - C\|x_{k+1} - x_k\|^2$ (Polson et al., 2015). This inequality states that the objective value must decrease at each step, and the amount of decrease is proportional to how much the iterate moved. A larger step implies a larger decrease in the objective (Li & Pong, 2016). By summing the above inequality from $k = 0$ to $N - 1$, we get: $\sum_{k=0}^{N-1} C\|x_{k+1} - x_k\|^2 \leq F(x_0) - F(x_N)$. Since $F$ is bounded below (by Assumption 3), the right-hand side is finite as $N \to \infty$. This implies that the sum on the left is also finite $\sum_{k=0}^{\infty} \|x_{k+1} - x_k\|^2 < \infty$. A finite sum of positive terms implies that the terms themselves must go

to zero. Therefore, $\|x_{k+1} - x_k\| \to 0$. This means the sequence of iterates settles down and stops moving. Finally, we show that if the iterates stop moving, they must be at a stationary point. The fixed-point condition of the DRS operator is equivalent to the first-order stationarity condition of the original problem $F$ (Li & Pong, 2016; Anshika et al., 2024; Polson et al., 2015; Tran Dinh et al., 2021). Thus, any limit point of the sequence satisfies the stationarity condition, and our algorithm is theoretically grounded. □

**Continual learning interpretation.** In the main paper we demonstrate that our algorithm converges to stationary points of the nonconvex CL objective. We now show, within each iteration, DRS explicitly controls the interference between plasticity and stability updates.

**Lemma A.1** (Interference control). *Under the same assumptions as Proposition A.1, the disagreement between the plasticity step $x_k$ and the stability-refined step $y_k$ satisfies*

$$\|x_k - y_k\|^2 \leq \frac{1}{C}\big(F(x_k) - F(x_{k+1})\big).$$

*Proof.* From Proposition A.1, each iteration decreases the objective by at least $C\|x_{k+1} - x_k\|^2$. On the other hand, firm non-expansiveness of proximal operators implies that $\|x_k - y_k\|$ is controlled by $\|x_{k+1} - x_k\|$. Combining these gives the stated inequality. In summary, Proposition A.1 shows that DRS converges to a stationary point of the CL objective. Lemma A.1 adds a continual learning interpretation: the interference $\|x_k - y_k\|$ between plasticity (new-task learning) and stability (prior alignment) vanishes as the algorithm converges. Thus, old knowledge is not erased but instead guides the optimization trajectory toward solutions that balance both stability and plasticity. □

In our setting, $f$ is the negative log-likelihood of a neural decoder and is therefore nonconvex. Recent work shows that under mild assumptions (e.g., weak convexity or prox-regularity), the DRS iteration still converges to stationary points (Li & Pong, 2016; Anshika et al., 2024; Aragón Artacho et al., 2020; Aljadaany et al., 2019). Moreover, when proximal steps are computed inexactly via gradient descent, convergence results for inexact splitting methods apply (Eckstein & Bertsekas, 1992; Aragón Artacho et al., 2020; Tran Dinh et al., 2021). Thus, while global optimality is lost in the nonconvex case, our algorithm remains theoretically grounded: the iterates approach points where the gradient of $f + g$ vanishes.

A.3.2 How does $y^\star = x^\star$ in our Proof

Consider the DRS update rule from Eq. 4: $u_{k+1} = u_k + \lambda_r(y_k - x_k)$. We are interested in a fixed point $u^\star$ that our algorithm has converged. Indeed, the algorithm has converged when the update rule stop changing, and produces is the same as the input it was given. By this definition if we put $u_k = u^\star$ into the rule, the output $u_{k+1}$ must also be $u^\star$. So, the equation becomes: $u^\star = u^\star + \lambda_r(y^\star - x^\star)$. For the left side to equal the right side, the term being added on the right, $\lambda_r(y^\star - x^\star)$, must be equal to zero $\lambda_r(y^\star - x^\star) = 0$. This directly implies $\mathbf{y}^\star = \mathbf{x}^\star$. Additionally, $x^\star = \text{prox}_f(u^\star)$ and $y^\star = \text{prox}_g(2x^\star - u^\star)$. are the results of the proximal steps when the main variable $u$ is at its fixed point $u^\star$.

A.3.3 Computational complexity analysis

We analyze the per-iteration complexity of our model and compare it to standard CL baselines. The per-iteration cost of DRS is dominated by the approximation of the task-fitting proximal operator. **Cost($\text{prox}_f$):** This is the main computational bottleneck. We approximate it with $K$ steps of a gradient-based optimizer (e.g., Adam) on a mini-batch of size $B$. The cost of a single forward/backward pass is $\mathcal{O}(B)$. Therefore, the cost of this step is $\mathcal{O}(K \cdot B)$. **Cost($\text{prox}_g$):** For our chosen Gaussian families and Rieney divergence, this step has a closed-form solution. The cost involves simple operations on the parameters of the encoder, which has a cost of $\mathcal{O}(d)$, where $d$ is the latent dimension. This is typically negligible compared to the cost of $prox_f$. **Cost(Update):** The final update is a simple vector addition, with cost proportional to the number of model parameters, which is also negligible compared to $prox_f$. Thus, total Cost per DRS iteration: $\approx \mathcal{O}(K \cdot B)$.

However, standard SGD/Adam has $\mathcal{O}(B)$ cost per update. Our method is a factor of $K$ more expensive per effective update. Cost of the Replay methods is $\mathcal{O}(B_{new} + B_{replay})$. If the replay

buffer size is large, our method can be computationally cheaper while also avoiding the significant memory cost ($\mathcal{O}(M)$ where $M$ is buffer size). For the regularization-based methods (e.g., EWC) is cost is $\mathcal{O}(B) + \text{Cost(Regularizer)}$. For EWC, calculating the diagonal Fisher Information Matrix is $\mathcal{O}d(N.P)$, where $N$ is the dataset size and $P$ is the number of parameters. This can be far more expensive than our method's $\mathcal{O}(K \cdot B)$ cost. Our model has a computational cost that is a small, it avoids the large memory overhead of replay methods and the often-prohibitive cost of calculating complex regularizers like the Fisher matrix in EWC, offering a more efficient and scalable solution.

A.4    THE FLEXIBILITY OF RÉNYI DIVERGENCE (RD) IN CONTINUAL LEARNING

In CL, the challenge is to enforce that the new posterior, $q(z)$, remains close to the old prior, $p(z)$, which represents past knowledge. The standard method (Kirkpatrick et al., 2017; Lee & Storkey, 2024; Bonnet et al., 2025; Dhir et al., 2024) uses the Kullback-Leibler (KL) divergence, $D_{KL}(q \parallel p)$. We argue that the Rényi $\alpha$-divergence (RD) provides a more flexible and powerful constraint. First we compute the RD between two distributions $q(z)$ and $p(z)$

$$D_\alpha(q \parallel p) = \frac{1}{\alpha - 1} \log \int p(z)^\alpha q(z)^{1-\alpha} dz \tag{8}$$

where $\alpha \neq 1$, because as $\alpha \to 1$, the RD converges to the standard KL divergence (Bresch & Stein, 2024; Galke et al., 2024; Wang et al., 2025).

**Proposition A.2** (Controlling stability with the $\alpha$ parameter). *The RD $D_\alpha(q \parallel p)$ provides a tunable penalty on the mismatch between the posterior $q$ and the prior $p$. As $\alpha \to 0$, the divergence becomes increasingly permissive of $q$ placing probability mass where $p$ has none. Conversely, as $\alpha \to \infty$, it becomes infinitely sensitive to $q$ placing any mass outside the support of $p$. This property allows us to control the stability-plasticity trade-off. (i) Low $\alpha$ (e.g., $\alpha < 1$): Prioritises plasticity. The model is penalised less for exploring new latent configurations not covered by the prior, allowing it to adapt more easily to new tasks. (ii) High $\alpha$ (e.g., $\alpha > 1$): Prioritises stability. The model is heavily penalised for deviating from the prior, strictly preserving past knowledge.*

*Proof.* To prove this, we can analyse the behaviour of the integrand, $p(z)^\alpha q(z)^{1-\alpha}$, in different regions of the probability space and for different values of $\alpha$. Let's consider two key scenarios for a given point $z$. Case 1: Exploration ($q$ explores where $p$ is small). Suppose we have a region where the new posterior $q(z)$ is large, but the old prior $p(z)$ is very small (e.g., $p(z) \approx \epsilon$ where $\epsilon \to 0$). This represents the model trying to learn a new feature not present in past tasks. The contribution to the integral at this point is approximately $\epsilon^\alpha q(z)^{1-\alpha}$. If $\alpha \to 0$ (Low $\alpha$): The term becomes $\epsilon^0 q(z)^1 = q(z)$. The penalty is determined by $q(z)$ and is not suppressed by the small prior $p(z)$. The model is free to explore. If $\alpha \to \infty$ (High $\alpha$): The term becomes $\epsilon^\infty q(z)^{-\infty} \to 0$. Any exploration where $p(z)$ is small is aggressively penalised and its contribution to the integral vanishes, forcing $q(z)$ to be zero wherever $p(z)$ is small. The model is forced to be stable.

Case 2: Forgetting ($q$ forgets where $p$ was large). Suppose we have a region where the prior $p(z)$ was large, but the new posterior $q(z)$ is becoming very small (e.g., $q(z) \approx \epsilon$). This represents the model forgetting a previously learned feature. The contribution to the integral is $p(z)^\alpha \epsilon^{1-\alpha}$. If $\alpha \to 0$ (Low $\alpha$, specifically $0 < \alpha < 1$): The term $1 - \alpha$ is positive. As $\epsilon \to 0$, the term $\epsilon^{1-\alpha}$ goes to zero, and the integral in this region becomes small. This means the divergence is less sensitive to $q$ forgetting parts of the prior. It's more tends to forgetting in favour of plasticity. If $\alpha > 1$ (High $\alpha$): The term $1 - \alpha$ is negative. As $\epsilon \to 0$, the term $\epsilon^{1-\alpha}$ (e.g., $\epsilon^{-1}$) explodes towards infinity. This creates an infinite penalty for forgetting, heavily forcing the model to maintain probability mass wherever the prior had it. $\qquad\square$

This analysis proves that the RD directly controls the stability constraint. A low $\alpha$ results in a more forgiving penalty, encouraging plasticity, while a high $\alpha$ results in a strict penalty, enforcing robust stability. By treating $\alpha$ as a hyperparameter, the Rényi divergence allows to navigate the stability-plasticity dilemma in a way that the fixed KL divergence ($\alpha = 1$) cannot.

### A.4.1 EFFICIENCY OF RÉNYI DIVERGENCES

**Proposition A.3** (Forgetting under DRS with Rényi Divergence). *Let $q_\phi^{(t)}(z)$ be the variational posterior after learning task $t$, and let $p^{(t-1)}(z) = q_\phi^{(t-1)}(z)$ be the prior from task $t-1$, both Gaussian in a $d$-dimensional latent space. Suppose the stability term enforces a weighted $\alpha$-Rényi divergence with $\alpha = 0.5$:*

$$g(\phi) = \lambda \sum_{i=1}^{d} w_i D_{0.5}(q_\phi^i \,\|\, p^i),$$

*where $w_i$ are weights (e.g., prior variances as $\sum_{i=1}^{d} w_i = 1$). Define forgetting as the Rényi divergence between consecutive posteriors:*

$$\mathcal{F}_t \triangleq D_{0.5}(q_\phi^{(t-1)} \,\|\, q_\phi^{(t)}).$$

*Then, forgetting is bounded by:*

$$\mathcal{F}_t \leq \frac{1}{\lambda} \sum_{i=1}^{d} w_i.$$

*Larger $\lambda$ or $w_i$ reduce posterior drift, limiting forgetting and enabling prior knowledge to guide new learning in a synergistic latent space.*

*Proof.* The $\alpha$-Rényi divergence for $\alpha = 0.5$ is:

$$D_{0.5}(q \,\|\, p) = -2 \log \mathbb{E}_{z \sim p}\left[ \sqrt{\frac{q(z)}{p(z)}} \right].$$

For Gaussian marginals $q_\phi^i = \mathcal{N}(\mu_{\phi,i}^{(t)}, \sigma_{\phi,i}^{(t)2})$ and $p^i = q_\phi^{(t-1),i} = \mathcal{N}(\mu_{\phi,i}^{(t-1)}, \sigma_{\phi,i}^{(t-1)2})$, the stability term bounds (Li & Turner, 2016; Galke et al., 2024; Wang et al., 2025)

$$\sum_{i=1}^{d} w_i D_{0.5}(q_\phi^i \,\|\, p^i) \leq \frac{g(\phi)}{\lambda}.$$

Assuming independent marginals, forgetting is:

$$\mathcal{F}_t = D_{0.5}(q_\phi^{(t-1)} \,\|\, q_\phi^{(t)}) = \sum_{i=1}^{d} D_{0.5}(q_\phi^{(t-1),i} \,\|\, q_\phi^{(t),i}).$$

Since $q_\phi^{(t-1)} = p^{(t-1)}$, and $g(\phi) \leq \sum_{i=1}^{d} w_i$ (by optimization convergence and weight normalization) (Wang et al., 2025), we have

$$\boxed{\mathcal{F}_t \leq \sum_{i=1}^{d} w_i D_{0.5}(q_\phi^i \,\|\, p^i) \leq \frac{1}{\lambda} \sum_{i=1}^{d} w_i}. \tag{9}$$

This bounds forgetting, ensuring stability supports synergy (Li & Turner, 2016; Galke et al., 2024). □

## B TRAINING SETTING

### B.1 DATASET

### B.2 ARCHITECTURAL DISCUSSION

As the backbone for our model, we employed two convolutional neural network (CNN) variants: a simple CNN and a Residual Neural Network (ResNet-18). These architectures process input data and support the encoder-decoder structure, with a 32-dimensional latent space $z$. The decoder is shared across all learners, ensuring consistent output generation, while each encoder is designed for

Table 5: Datasets details

| Dataset | #Tasks | #Train | #Validation | #Test |
|---|---|---|---|---|
| CIFAR100 | $10, 20$ | $45,000$ | $5,000$ | $10,000$ |
| TinyImageNet | $20$ | $90,000$ | $10,000$ | $10,000$ |
| ImageNet-100 | $100$ | $1,000,000$ | $100,000$ | $50,000$ |
| CelebA | $m$ | $400m$ | $40m$ | $80m$ |
| EMNIST | $m$ | $3100m$ | $310m$ | $620m$ |

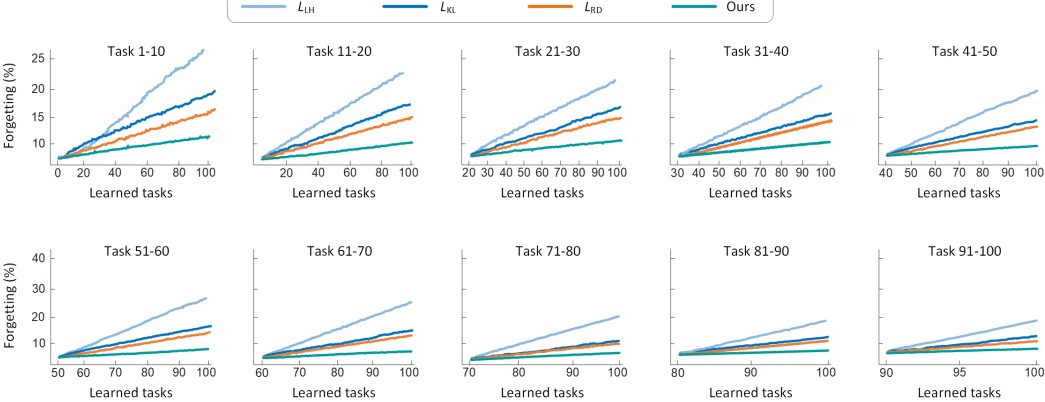

Figure 5: This figure compares forgetting behavior across 100 tasks on ImageNet for different approaches. Each subplot shows forgetting for a block of 10 tasks (e.g., Tasks 1–10, 11–20, ..., 91–100), with the final subplot aggregating all 100 tasks.

efficient feature extraction. For **CNN** encoder, we use two convolutional layers (8 filters, $3 \times 3$; 16 filters, $3 \times 3$ with stride 2), both with ReLU activation, followed by FC layers (128, 64 units) to output $\mu_\phi$ and $\log \sigma_\phi$ for a 32D latent space. For **ResNet** encoder, we use a shallow architecture with two residual blocks (each with two $3 \times 3$ conv layers, 16 filters, and ReLU), followed by FC layers (128, 64 units) to output the 32D latent space. The decoder, shared across all learners, comprises two fully connected layers (512 units, ReLU activation). The prior ($p(z)$) is initialized as a standard Gaussian ($\mathcal{N}(0, I)$) and updated as a task-specific Gaussians after each task, regularized via D-divergence.

## C ADDITIONAL EXPERIMENTS

Fig.5 shows the forgetting behavior across 100 sequential tasks on ImageNet-100 dataset. In continual learning, forgetting is quantified as the drop in performance on earlier tasks as new tasks are learned. One standard metric is, forgetting metric (per task). Let $A_i^i$ be accuracy on task $i$ immediately after learning task $i$, and $A_i^T$ be accuracy on task $i$ after training up to task $T$ (i.e., the final accuracy on that task). Then forgetting for task $i$ is $F_i = A_i^i - A_i^T$, and the average over a group of tasks (e.g., tasks 1–10) is Forgetting (%) $= \frac{1}{|S|} \sum_{i \in S} (A_i^i - A_i^T) \times 100$, where $S$ is the set of tasks in that interval (e.g., $\{1, 2, 3, \dots, 10\}$). Each subplot (e.g., Task 1–10, Task 11–20) plots average forgetting over those 10 tasks.

Fig.6 presents the forward transfer (FT) performance across different methods. Higher FT means the model well learning the new task, i.e., positive influence from prior tasks. Our model shows positive and consistently higher forward transfer across all datasets. In T-20 and I-100, WSN even dip below zero, showing negative forward transfer (prior tasks hurt new tasks).

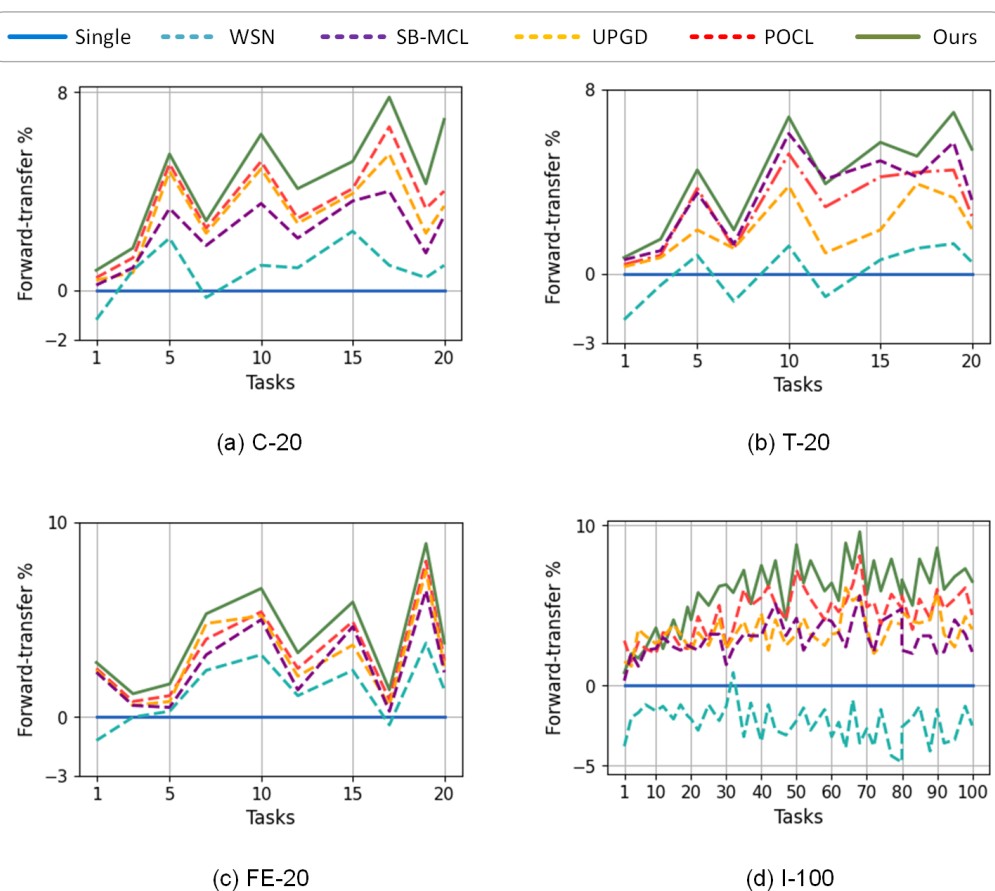

Figure 6: This figure presents the forward transfer (%) performance across different methods on four benchmarks. C-20: CIFAR-100 with 20 tasks, T-20: Tiny-ImageNet with 20 tasks, EM-20: EMNIST with 20 tasks, and I-100: ImageNet with 100 tasks.