# OpenReview forum: "Finding Structure in Continual Learning"
_ICLR.cc/2026/Conference — ICLR 2026 Conference Withdrawn Submission_

### Official Review · Reviewer_nf3M · 2025-10-27

**Soundness:** 2
**Presentation:** 1
**Contribution:** 2
**Rating:** 2
**Confidence:** 4

**Summary:**

The paper presents a novel approach to continual learning using Douglas-Rachford Splitting (DRS), aiming to address the stability-plasticity dilemma. This approach reformulates the learning process as a negotiation between two decoupled objectives, which helps to integrate new information while retaining previous knowledge. The efficacy of the method is demonstrated through experiments on different datasets such as EMNIST, CIFAR-10/100, ImageNet, and TinyImageNet, along with various neural network architectures.

**Strengths:**

1. Introducing DRS into continual learning seems to be novel.

2. The method was tested across a wide range of datasets, underlining its robustness and broad applicability.

3. Demonstrates substantial improvements in minimizing forgetting and enhancing knowledge transfer, making notable progress compared to traditional approaches.

**Weaknesses:**

1. The paper is poorly written:
    * In the proof for the proposition, the authors just cited a lot of previous works, using their results. It's hard for the reviewer to verify the correctness.
    * The paper lists a lot of baselines without indicating what specific methods they use. It's hence hard to verify why the proposed method is superior.

2. Lack of Sequential Meta-Learning Comparison: Given that the proposed approach resembles sequential meta-learning, comparisons with similar Bayesian meta-learning methods are lacking. Important related works such as sequential Bayes meta-learning should have been included  [1,2].

[1] Matthew Riemer, Ignacio Cases, Robert Ajemian, Miao Liu, Irina Rish, Yuhai Tu, and Gerald Tesauro. Learning to learn without forgetting by maximizing transfer and minimizing interference. arXiv preprint arXiv:1810.11910, 2018.

[2] Xu He, Jakub Sygnowski, Alexandre Galashov, Andrei A Rusu, Yee Whye Teh, and Razvan Pascanu. Task agnostic continual learning via meta learning. arXiv preprint arXiv:1906.05201, 2019.

**Questions:**

1. What structure have you found? I cannot get the meaning of the paper title.

2. When the tasks change significantly compared to previous tasks, does the objective still make sense?

3. What are the exact benefits of DRS? Converging to a stationary point does not suggest a good generalization over all the tasks, while it works for the empirical objective. How would the algorithm like of each task only contain a few samples?

---

### Official Review · Reviewer_9pjk · 2025-10-29

**Soundness:** 2
**Presentation:** 3
**Contribution:** 2
**Rating:** 4
**Confidence:** 4

**Summary:**

This paper proposes a continual learning approach based on Douglas-Rachford Splitting (DRS) to address the stability-plasticity dilemma. The authors argue that the core issue in continual learning is not the objective itself but the optimization strategy that forces competing objectives into direct conflict. They employ DRS to decouple task-fitting (plasticity) and prior-alignment (stability) terms through separate proximal operators, combined with Rényi divergence for the regularization term. Experiments on multiple benchmarks show improvements over baseline methods.

**Strengths:**

1. The biggest advantage of this paper is that it has minimal storage overhead while maintaining high performance. The method is entirely replay-free and doesn't require additional model components, making it practical for resource-constrained settings.

2. The paper offers a fresh perspective by reframing continual learning as an optimization problem rather than an objective design problem. This aligns with recent work [1] showing that how we optimize can be as important as what we optimize.

3. The theoretical framework provides convergence guarantees for the DRS-based optimization.

[1] A Unified and General Framework for Continual Learning, ICLR 2024

**Weaknesses:**

1. My biggest concern is the inconsistency of the motivation. The paper's central claim that optimization strategy is the fundamental problem in CL needs stronger validation. While we acknowledge that DRS and Rényi divergence might work best together, the paper doesn't show why this combination is necessary. At minimum, testing DRS with KL divergence would help us understand whether the improvements come from the optimization strategy or the divergence choice. Though the authors focus on replay-free methods, regularization-based approaches like EWC are also replay-free and would be ideal for testing the generality of DRS.

2. By decoupling the objectives through separate proximal operators, the method might miss beneficial interactions between plasticity and stability. In joint optimization, gradients can reinforce each other when they align, potentially enabling positive transfer. The paper claims DRS enables synergistic learning but doesn't provide evidence that decoupled optimization preserves these beneficial interactions.

3. The simultaneous change of multiple components makes it hard to assess individual contributions. We understand computational constraints may limit extensive ablations, but even one experiment applying DRS to an existing regularization method would strengthen the claims significantly. [1] shows that various CL methods share common mathematical structures, suggesting DRS could potentially improve them if it addresses a fundamental optimization issue.

4. The paper doesn't clearly position itself relative to other optimization-focused CL approaches like gradient projection[2] or bi-level optimization[3] or loss landscape methods [4][5]. How does DRS compare empirically to such approaches?

[1] A Unified and General Framework for Continual Learning.

[2] Gradient projection memory for continual learning.

[3] Look-ahead meta learning for continual learning.

[4] Linear mode connectivity in multitask and continual learning.

[5] Make Continual Learning Stronger via C-Flat.

**Questions:**

1. What values of K (gradient steps for approximating prox_f) were used in practice? How sensitive is performance to this choice?

2. Could you test DRS on one regularization-based baseline like EWC? This maintains your replay-free constraint while testing generality.

---

### Official Review · Reviewer_nUFo · 2025-11-01

**Soundness:** 2
**Presentation:** 2
**Contribution:** 3
**Rating:** 2
**Confidence:** 4

**Summary:**

The paper proposes using Douglas–Rachford Splitting (DRS) to optimize continual learning objectives by decoupling plasticity and stability terms. The idea is to replace the usual joint gradient descent on f+g with alternating proximal updates. The method combines DRS with a Bayesian latent-space model and Rényi divergence, reporting strong results on several benchmarks.

**Strengths:**

- The theoretical background is solid, and the motivation of leveraging DRS for a composite objective makes sense.
- Clear formulation and well-written exposition of the theory.
- Large number of comprehensive experiments with a significant number of baselines.
- The method is replay-free and only optimization-based

**Weaknesses:**

## Weaknesses
1. Some claims regarding the impact of DRS are not significantly backed up or overstated. DRS is a generic solver for any regularized objective; hence, every regularization-based CL method can be reformulated similarly. Experiments replacing the SGD optimizer with DRS for existing regularization-based methods would be appreciated to showcase the effectiveness of DRS.
2 The paper combines the optimizer DRS with other changes (Rényi divergence, Bayesian latent models, tuned hyperparameters). The contribution of DRS itself is not isolated. Meanwhile, the paper claims "our contribution is an optimization scheme based on Douglas-Rachford Splitting", l.193. I strongly advise the authors to conduct an ablation study that demonstrates the impact of DRS alone, especially when the usage of Rényi seems to be the most significant here.
3. Following previous points, the experimental comparison is misaligned: results compare DRS + Rényi + Bayesian model to heterogeneous CL baselines instead of a clean SGD vs DRS study on the same objective.
4. The hyperparameter selection is not detailed. Hyperparameter sensitivity is key in Continual Learning. Given the current gain in performance, clarifying this point is crucial.
5. The divergence parameter $\alpha$ is never defined clearly in the paper. Also, when reading the ablation study section (4.3), it seems that this parameter has the most impact on the stability plasticity trade-off. This paper should include a demonstration of the effect of DRS **alone**, as announced in the introduction. Showing that DRS can reduce forgetting, as in Figure 3, is anecdotal.
6. Eventually, I found the current exposition rather confusing. The paper argues that the main contribution is the optimization strategy, but it compares mostly to loss functions rather than optimization strategies.

Overall, the formulation is interesting, but I believe the current evaluation is unclear. Given that the authors can address my concerns I would happily raise my score.

## Typos
- l.130: exiting
- In the abstract, the sentence "Most methods address this by summing competing loss terms, creating gradient conflicts that are managed with complex and often inefficient strategies such as external memory replay or parameter regularization." is confusing to me. Memory does not include the competing loss term, as the loss is unchanged, only the data. Memory methods are orthogonal to regularization-based methods, and this paper tackles regularization, memory-free methods (but could certainly be used with a memory buffer). I would rephrase this to avoid confusion.

**Questions:**

See above

---

### Official Review · Reviewer_eMUA · 2025-11-02

**Soundness:** 2
**Presentation:** 3
**Contribution:** 2
**Rating:** 4
**Confidence:** 4

**Summary:**

The authors propose a continual learning optimizer based on Douglas–Rachford Splitting. The way they apply DRS is interesting/unusual: there is the plasticity step (which us standard stochastic likelihood updates) and the stability step (enforcing prior-posterior consistency through a weighted Renyi divergence). The encoder parameters are updated in both steps, while the decoder is optimized only during plasticity. The authors claim that this alternating process implicitly performs Bayesian inference, yielding convergence to stationary points and mitigating catastrophic forgetting without replay. Experiments on vision benchmarks report improved average accuracy and backward transfer compared to regularization-based baselines. However, the theoretical contributions remain underdeveloped: the propositions rely on convex or exact proximal assumptions that do not hold in the proposed nonconvex, stochastic setting. Also the benefits of the Renyi term over KL divergence are only argued heuristically. As a result, the framework is conceptually interesting but mathematically immature and empirically incremental relative to prior proximal or Bayesian continual-learning methods.

**Strengths:**

+ Introduces a conceptually novel connection between DRS and CL. I have not seen the application of operator splitting methods in continual learning before
+ The alternation between plasticity (data likelihood) and stability (Renyi prior alignment) provides an interpretable framework linking optimization dynamics to approximate Bayesian inference
+ The use of Renyi divergence as a stability regularizer is something new I think and theoretically motivated by avoiding the KL zero-forcing bias.
+ The algorithm is replay-free and memory-efficient, addressing a common limitation of many CL methods.
+Empirical evaluation spans multiple standard datasets and metrics (ACC, BWT, FWT), showing consistent though moderate performance gains over standard regularization baselines.
+ The presentation is clear and well structured, with pseudocode, ablations, and hyper-parameter sensitivity analyses that improve reproducibility

**Weaknesses:**

1. The theoretical analysis is incomplete. The propositions rely on convex assumptions and exact proximal updates while the actual algorithm is nonconvex and inexact, invalidating the stated convergence guarantees.
2. Prop-3.1 incorrectly argues that the KL divergence causes “zero forcing” in Gaussian families. It fails to formally derive the claimed advantage of the Renyi term
3. Prop-3.2 applies Douglas–Rachford results without addressing error accumulation from the K-step SGD approximation to the proximal operator
4. there is no proof or citation to support convergence under the paper’s nonconvex stochastic regime. The discussion in Appendix A is mostly a speculation.
5. The Bayesian interpretation (approximate posterior sequence) sounds intuitive but it is not formally derived from a probabilistic model or variational objective
6. Empirical improvements over prior methods are small and limited to few datasets. There is no comparison to recent proximal (e.g., LPR) or Bayesian (e.g., BLCL) baselines.

**Questions:**

I hope that the following suggestions will help improve the paper:

- plz rework the theoretical section. First, I suggest you define all assumptions (convexity, smoothness, exact vs. inexact prox) and restate the propositions with mathematically valid guarantees. If proximal updates are approximate, use an inexact DRS framework and maybe prove epsilon-stationarity rather than exact convergence
- replace the informal KL zero-forcing argument with a closed-form comparison of Gaussian–Gaussian KL and Renyi divergence. You can show gradients with respect to the mean and covariance for the alpha values used in experiments
- plz clarify whether the stability proximal update applies only to the encoder and analyze its effect on likelihood calibration, negative log-likelihood, or feature drift relative to the decoder
- maybe you can have a Lyapunov or Douglas-Rachord envelope descent argument (or citation) for non-convex DRS to substantiate Proposition 3.2
- Plz include standard deviations or confidence intervals for all reported metrics. I suggest you add shaded uncertainty bands in figures and explain how variability is computed
- It is important to compare against stronger baselines. For example Layerwise Proximal Replay (Yoo et al, ICML 2024) and recent Bayesian continual-learning methods,
- plz conduct sensitivity analyses for alpha, lambda, gamma -- to evaluate robustness

---

### Note · Authors · 2025-11-22

I have read and agree with the venue's withdrawal policy on behalf of myself and my co-authors.